# Dental developmental complications in pediatric hematopoietic stem cell transplantation patients: A study using CMC clinical data warehouse

**Jaehyun Kim[1], Hee Jin Lim[1], Ja Hyeong Ku[1], Yoon-Ah Kook[1], Nack-Gyun Chung[2], Yoonji Kim[1] ***

**1** Department of Orthodontics, Seoul Saint Mary's Hospital, College of Medicine, The Catholic University of Korea, Seoul, Korea, **2** Department of Pediatrics, Seoul Saint Mary's Hospital, College of Medicine, The Catholic University of Korea, Seoul, Korea

* juice@catholic.ac.kr, kyoonji@gmail.com

**Data Availability Statement:** All data are available from a public repository (Kim J, Lim HJ, Ku JH, Kook YA, Chung NG, Kim Y. Dental developmental complications in pediatric hematopoietic stem cell

## Abstract

### Objective

This study aimed to investigate the prevalence and extent of dental developmental complications in patients who have undergone pediatric hematopoietic stem cell transplantation (SCT) and identify the risk factors.

### Materials and methods

We retrospectively investigated the clinical data warehouse of the Catholic Medical Center information system for identifying patients who: 1) visited the Department of Pediatrics between 2009 and 2019, 2) underwent SCT under the age of 10, and 3) had panoramic radiographs. Thus 153 patients were included in this study. The prevalence and extent of tooth agenesis, microdontia, and root malformation were assessed using panoramic radiographs obtained after SCT, and the risk factors were analyzed using regression analysis.

### Results

All 153 patients had at least one dental anomaly. When grouped according to the age at initial chemotherapy ($\leq$ 2.5; 2.6–5.0; 5.1–7.5; > 7.5 years), the prevalence of agenesis showed statistically significant differences among the different age groups (P < 0.001). The prevalence of agenesis was highest in the youngest age group. As the initial age at chemotherapy increased, the number of affected teeth per patient decreased for all three anomalies. The location of the affected tooth was also influenced by the age at initial chemotherapy. Regression analysis demonstrated that young age at initial chemotherapy was a risk-increasing factor for tooth agenesis and microdontia.

transplantation patients: A study using CMC clinical data warehouse. 2022. Figshare Digital Repository. https://doi.org/10.6084/m9.figshare.21397647).

**Funding:** This study was supported by Seoul St. Mary's Hospital, The Catholic University of Korea, Seoul, Korea (ZC20RISI0730 / https://www.cmcseoul.or.kr/en.common.main.main.sp) and National Research Found government (MSIT) (No. 2021R1F1A1049557 / https://www.nrf.re.kr/eng/main/). YK received all award above. All of these funding sources had no role in the design of this study and will not have any role during its execution, analyses, interpretation of the data, preparation of the manuscript, or decision to submit results. There was no additional external funding received for this study.

**Competing interests:** The authors have declared that no competing interests exist.

## Conclusions

The age at initial chemotherapy may be a critical factor in determining the type, extent, and location of dental complications after SCT. These results suggest that careful dental follow-up and timely treatment are recommended for pediatric patients undergoing SCT.

## Introduction

As the survival rate of childhood cancer patients increases, improving the quality of life (QOL) has become an essential task for cancer therapy [1, 2]. Few survivors are free of long-term side effects, and follow-up for these patients is essential for maintaining their QOL. Hematopoietic stem cell transplantation (SCT) plays a critical role in the treatment of malignancies in pediatric patients. Disturbances in dental development have been known to occur after SCT combined with chemotherapy and/or total body irradiation (TBI) in patients with childhood cancer [3–6]. Dental complications are not life-threatening; however, they may have a profound impact on QOL.

Developmental dental anomalies include tooth agenesis, microdontia (small-sized tooth), impaired and arrested root development, such as short V-shaped roots, and enamel hypoplasia [7–10]. These anomalies are characterized by defective hard tissue formation. Näsman et al. reported that all developing teeth were affected by multi-agent chemotherapy and radiation therapy [11] and that dental sequelae are irreversible. In addition, impaired salivary function [12], increased risk of dental caries, disturbances in craniofacial growth, and higher risk of developing secondary oral tumors have also been reported [3]. The prevalence of dental anomalies among pediatric patients with cancer is relatively high at 55.5%, compared with the prevalence of 6.7% among healthy individuals [13, 14]. Other studies have reported that the prevalence of dental anomalies ranged from 50–100% in patients who underwent SCT [15] and that disturbances in root development were present in all patients who underwent SCT [7].

A large amount of long-term data is required to evaluate late dental complications after SCT [16]. Most studies analyzed only specific factors in a limited numbers of samples [7, 17–19]. In addition, there are conflicting reports suggesting that dental disturbances can be attributed to chemotherapy regimens [20], the age of the patient and the use of cranial radiation therapy [11], or the effect of TBI [18]. From our clinical experience of treating a large number of adult and pediatric patients who underwent SCT, we postulated that developmental dental complications may only occur in growing individuals who have developing teeth, and the types and extent of dental complications after SCT may depend on the timing of anti-cancer treatment and/or on the treatment regimen.

To determine the treatment methods responsible for such a wide range of variables, large samples of pediatric SCT patients with dental follow-up data are required. The concept of big data from a clinical data warehouse (CDW) was used in this study to optimize large population selection and data analysis.

### Aim

This study aimed to investigate the prevalence and extent of developmental dental complications, including tooth agenesis, microdontia, and root malformation, after undergoing pediatric hematopoietic SCT using CDW data. The null hypothesis was that the timing of chemotherapy and TBI would not affect dental developmental complications.

## Materials and methods

### Study design and data collection

We retrospectively investigated the CDW of the Catholic Medical Center information system (CMC nU) to identify patients who meet the following inclusion criteria: 1) patients who visited the Department of Pediatrics at Seoul St. Mary's Hospital, The Catholic University of Korea, between January 2009 and December 2019, 2) underwent hematopoietic stem cell transplantation (SCT) under the age of 10, and 3) had panoramic dental radiographs taken.

In total, 207 patients were identified using the CDW data. Fifty-one patients were excluded because panoramic radiographs taken after SCT were not available. Two patients were excluded due to the poor quality of the panoramic radiographs, and one patient was excluded due to inaccurate medical history at other hospitals. Thus, 153 eligible patients were included in this study. A flowchart for the patient selection process is shown in Fig 1. The medical and demographic information of the patients, including age, sex, diagnosis at the time of SCT, types of chemotherapy agents, age at chemotherapy, and total body irradiation (TBI), were obtained from the CDW data. The experimental protocols were approved by the institutional review board of The Catholic University of Korea (KC20WASI0215).

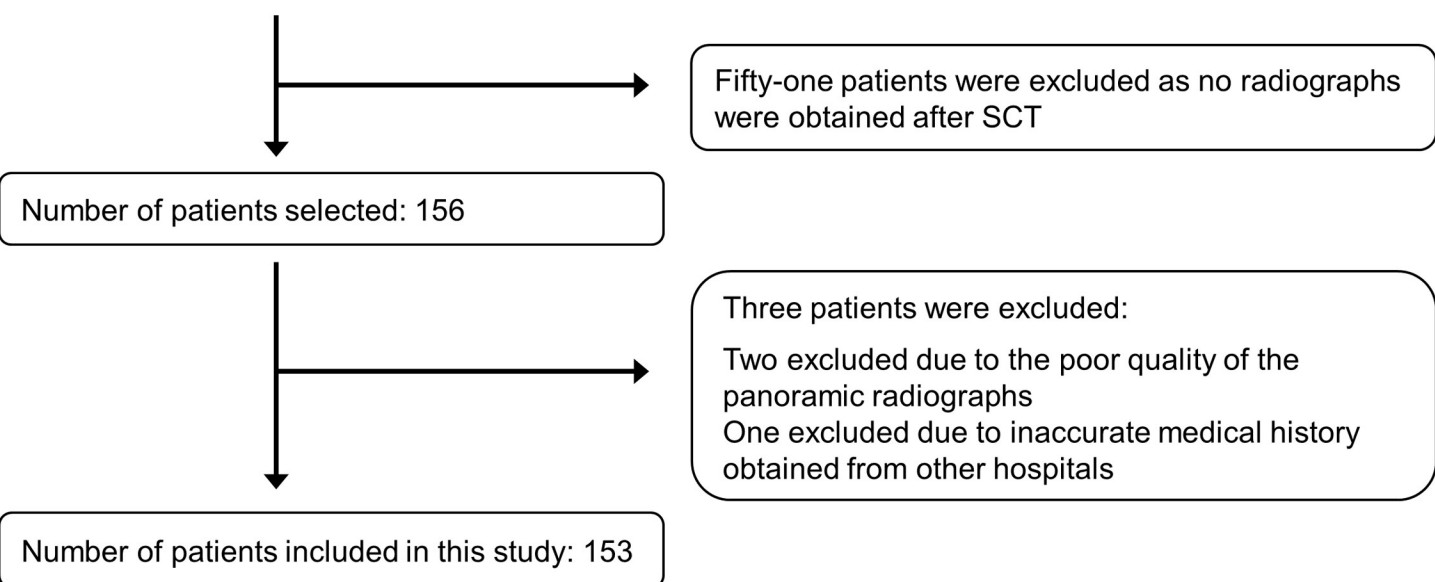

**Fig 1. Flowchart for study sample selection.**

## Identification of dental developmental anomalies using panoramic radiographs

Tooth agenesis, microdontia, and root malformation were examined using panoramic dental radiographs obtained after undergoing SCT (Fig 2). Agenesis of the permanent teeth indicated the congenital absence of teeth due to failure of tooth formation. Microdontia was defined as a tooth that is smaller in size than the normal tooth on the contralateral side comparing the mesiodistal width of the crown [21]. Root malformation was detected when a poor crown/root ratio or slender or short roots were observed.

## Study design

The incidence, prevalence, and extent of these dental anomalies were assessed among different age groups. The percentage of patients affected by each dental anomaly was defined as prevalence of each dental anomaly, and the percentage of affected teeth per patient was prescribed as incidence of each dental anomaly. The patients were grouped according to the initial age at chemotherapy ($\leq$ 2.5 years; 2.6–5.0 years; 5.1–7.5 years; > 7.5 years). In patients with tooth agenesis, the total number of permanent teeth may vary from the normal value of 28. Therefore, the incidence was calculated as the mean number of teeth affected per patient's total permanent teeth, excluding the congenitally missing teeth. Third molars and deciduous teeth were not included in the assessment. To investigate the risk factors for these developmental dental complications, the patients were divided into TBI and non-TBI groups, and the incidences of dental anomalies were compared. Regression analysis was used to assess the potential clinical risk factors associated with the three developmental dental anomalies.

## Statistical analysis

The prevalence and incidence of dental anomalies according to the initial age at chemotherapy were compared using the chi-squared test. The incidence of dental anomalies according to TBI was analyzed using Student's $t$-test. The outcome measure of agenesis, microdontia, and root malformation were analyzed using univariate logistic regression analysis to assess the risk factors. Multivariate logistic regression analysis was performed subsequently including the variables with $P < 0.15$ from the univariate regression to adjust for possible confounders. Statistical significance was set at $P < 0.05$ and all statistical analyses were performed using SPSS version 21 (IBM Corp., Armonk, NY, USA).

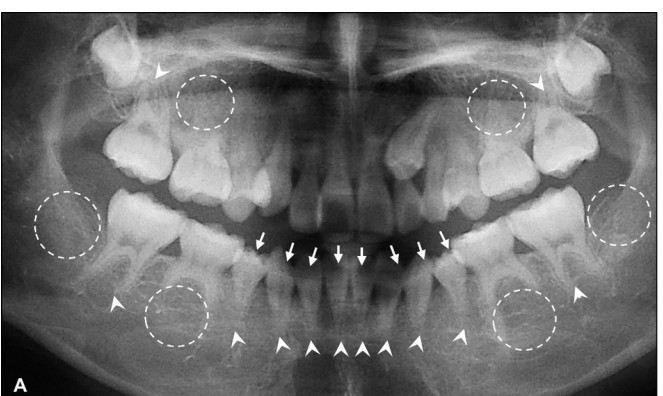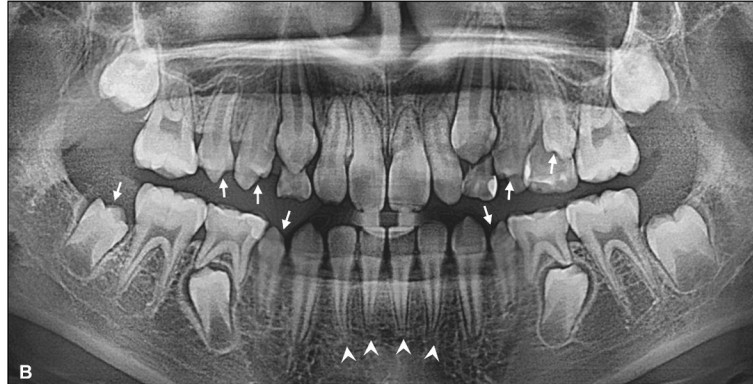

**Fig 2. Examples of dental anomalies found in the panoramic radiographs.** (A) and (B), Agenesis of permanent teeth (*dotted circle*), microdontia (small-sized teeth; *solid arrow*), and root malformation (*arrowhead*).

# Results

## Characteristics of the patients and conditioning before SCT

Table 1 shows the key characteristics and conditioning before undergoing SCT of the study patients based on the medical history. Acute lymphoblastic leukemia (ALL) and acute myeloid leukemia (AML) were the most common malignancies. The mean age at the time of initial chemotherapy of the patients was lower than that of patients with TBI. Approximately half of the patients underwent TBI.

## Prevalence of dental anomalies according to the initial age of chemotherapy

All patients (100%) had at least one dental anomaly on the panoramic radiographs. Tooth agenesis was found in 22.2% of patients, microdontia in 94.8%, and root malformation in 99.3% shown in supporting dataset. The patients were grouped according to the initial age at chemotherapy ($\leq$ 2.5 years; 2.6–5.0 years; 5.1–7.5 years; > 7.5 years). The percentage of patients affected by each dental anomaly is described in Table 2 and Fig 3A. The prevalence of agenesis showed a statistically significant difference among the different groups (P < 0.001), and the prevalence was the highest in the youngest age group. The prevalence of microdontia and root malformations did not show a statistically significant difference among the different age groups.

## Incidence of dental anomalies according to the initial age at chemotherapy

The incidence of dental anomalies was showed in Table 3 and Fig 3B. The incidence of microdontia and root malformation showed a statistically significant difference among the different groups. (P < 0.001). As the initial age at chemotherapy increased, the incidence of agenesis declined markedly. In the youngest group (age $\leq$ 2.5 years), the mean number of congenitally absent teeth was 5.1, whereas the mean number was 1.7 in the 2.6–5.0 age group. None of the patients in the oldest group (age > 7.5 years) had tooth agenesis. Microdontia and root malformation also showed the same tendency of having a smaller number of affected teeth in the older age groups.

## Distribution of dental anomalies according to the tooth location

The location of the affected tooth was influenced by the age at chemotherapy depending on the developmental stage of each tooth. The most common tooth exhibiting agenesis was the maxillary second premolar (U5; 25.6%), followed by the mandibular second premolar (L5; 22.7%). The incisors (U1-2, L1-2), canines (U3, L3), and first molars (U6, L6) were minimally affected by agenesis and microdontia. Microdontia was most prevalent in the mandibular first premolars (L4; 12.7%) and mandibular second premolars (L5; 12.0%). Root malformation was the most prevalent in the mandibular first premolar (L4; 9.5%) and maxillary first premolar (U4; 9.2%).

The distribution of each dental anomaly according to the location of the tooth in the different groups is shown in Fig 4. Microdontia mainly affected the anterior teeth (U1-3, L1-3) in the younger age groups, whereas it mainly affected the premolar (U4-5, L4-5) and second molars (U7, L7) in the older age groups (Fig 4B). Root malformation mainly affected the anterior teeth (U1-3, L1-3) and first molars (U6, L6) in the younger groups, whereas it mainly affected the premolars (U4-5, L4-5) and second molars (U7, L7) in the older age groups (Fig 4C).

**Table 1. Characteristics of the study patients and conditioning before undergoing SCT.**

| Diagnosis | Total | Chemotherapy | | | | | | | | | TBI | | Mean age at initial chemotherapy | Mean age at TBI |
|---|---|---|---|---|---|---|---|---|---|---|---|---|---|---|
| | | Cyclophosphamide | Fludarabine | Busulfan | Carboplatin | Thiotepa | Cytarabine | Etoposide | Melphalan | Carmustine | Yes | No | | |
| ALL | 49 | 48 | 19 | 13 | 31 | 7 | 29 | | | | 44 | 5 | 6.1 | 7.6 |
| AML | 36 | 6 | 36 | 35 | 24 | 25 | 16 | | | 1 | 17 | 19 | 5.0 | 7.2 |
| SAA | 17 | 17 | 17 | | | | | | | | 0 | 17 | 7.9 | |
| JMML | 8 | 3 | 7 | 8 | 3 | 2 | 1 | | | | 2 | 6 | 4.7 | 6.8 |
| MDS | 9 | 4 | 9 | 6 | 1 | | | | | | 1 | 8 | 7.4 | 8.3 |
| HLH/LCH | 4 | 4 | 1 | 3 | | 4 | | | | | 0 | 4 | 2.2 | |
| Lymphoma | 2 | 1 | | | | 2 | | 1 | | | 1 | 1 | 9.7 | 10.5 |
| Other bone marrow failure disorder | 13 | 12 | 8 | 6 | | | | | | | 2 | 11 | 7.1 | 8.7 |
| Other solid tumors | 15 | 15 | | | | 15 | | 15 | 9 | | 3 | 12 | 3.6 | 4.1 |
| Total | 153 | 110 | 97 | 71 | 59 | 55 | 46 | 16 | 9 | 1 | 70 | 83 | 5.8 | 7.4 |

SCT, stem cell transplantation; TBI, total body irradiation; ALL, acute lymphoblastic leukemia; AML, acute myeloid leukemia; SAA, severe aplastic anemia; JMML, juvenile myelomonocytic leukemia; MDS, myelodysplastic syndrome; HLH, hemophagocytic lymphohistiocytosis; LCH, Langerhans cell histiocytosis; mean age in years.

**Table 2. Prevalence of dental anomalies according to the age at initial chemotherapy.**

| Number (%) of patients | Age at initial chemotherapy (yrs) | | | | | | | | P-value |
|---|---|---|---|---|---|---|---|---|---|
| | ≤ 2.5 | | 2.6–5.0 | | 5.1–7.5 | | > 7.5 | | |
| Agenesis | 19 | (73.1) | 12 | (30.8) | 3 | (7.9) | 0 | (0.0) | 0.0000*** |
| Microdontia | 26 | (100.0) | 38 | (97.4) | 37 | (97.4) | 44 | (88.0) | NS |
| Root malformation | 26 | (100.0) | 39 | (100.0) | 38 | (100.0) | 49 | (98.0) | NS |
| Total | 26 | | 39 | | 38 | | 50 | | |

Chi-square test. NS indicates no statistical significance between the groups.

*P < .05

**P < .01

***P < .001.

## Incidence of dental anomalies according to TBI

Table 4 shows that the incidence of agenesis was significantly higher in the non-TBI group than that in the TBI group in the youngest age group (P < 0.05). In the case of microdontia and root malformation, there were no significant differences between the TBI and non-TBI groups, except for microdontia in the 2.6–5.0 age group.

## Multivariate regression analysis on the risk factors for developmental dental complications

The age at initial chemotherapy, presence of TBI, age at TBI, diagnosis, and the number of chemotherapy agents were selected as risk factors shown in Table 5. Initially, the univariate

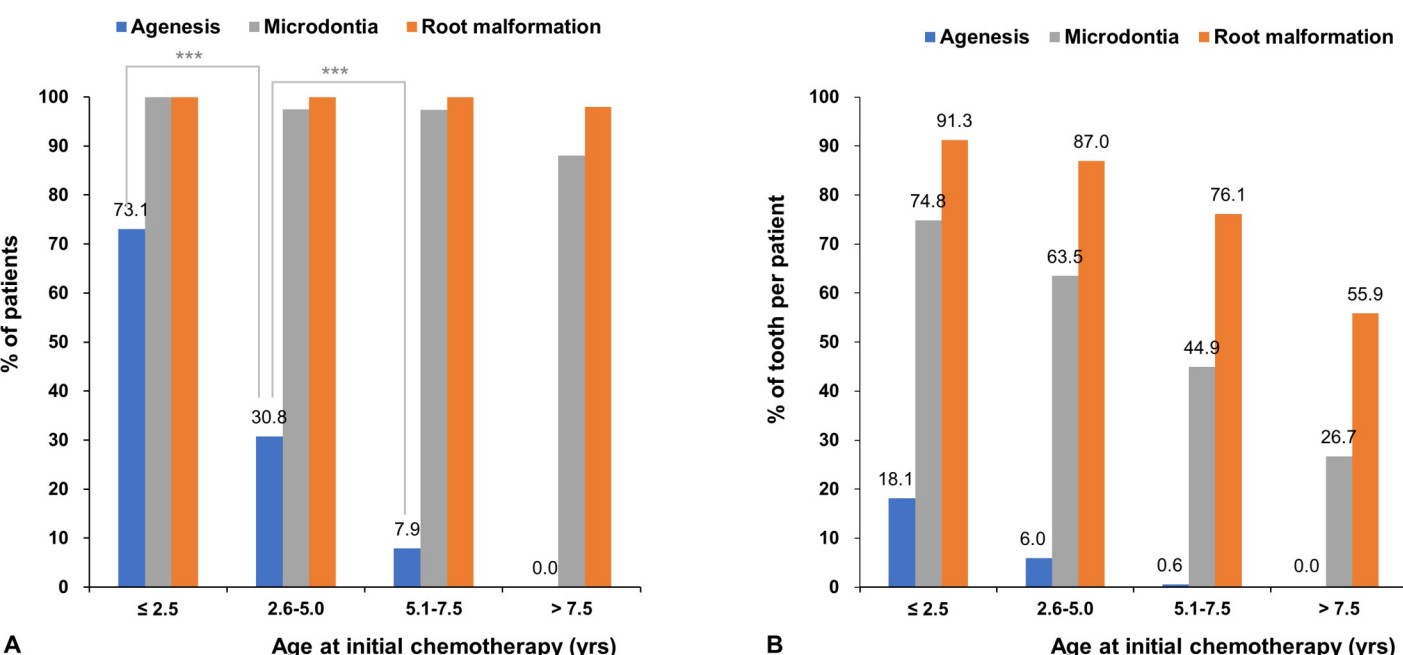

**Fig 3. Prevalence and incidence of dental anomalies according to the initial age at chemotherapy.** (A) The prevalence of agenesis showed a statistically significant difference among the different groups (P < 0.001), and the prevalence is highest in the youngest age group. The prevalence of microdontia and root malformation do now show a statistically significant difference among the groups. Chi-square test. ***P < 0.001 (B) The incidence of microdontia and root malformation show a statistically significant difference between the groups. As the initial age at chemotherapy increases, the incidence of agenesis declines markedly. In the oldest group of > 7.5 years, there are no patients with tooth agenesis. Microdontia and root malformation also show the same tendency of a smaller number of affected teeth among patients who were older at initial chemotherapy.

**Table 3. Incidence of dental anomalies according to the initial age at chemotherapy.**

| Mean number (%) of tooth per patient | Age at initial chemotherapy (yrs) | | | | | | | | P-value |
|---|---|---|---|---|---|---|---|---|---|
| | ≤ 2.5 | | 2.6–5.0 | | 5.1–7.5 | | > 7.5 | | |
| Agenesis | 5.1 | (18.1) | 1.7 | (6.0) | 0.2 | (0.6) | 0.0 | (0.0) | 0.0000*** |
| Microdontia | 17.0 | (74.8) | 16.4 | (63.5) | 12.5 | (44.9) | 7.5 | (26.7) | 0.0000*** |
| Root malformation | 20.8 | (91.3) | 22.7 | (87.0) | 21.2 | (76.1) | 15.6 | (55.9) | 0.0000*** |

Chi-square test. NS indicates no statistical significance between the groups.

*P < .05

**P < .01

***P < .001.

analysis for agenesis was performed and the age at initial chemotherapy, TBI, diagnosis of other solid tumors, JMML and ALL had values of P < 0.15. Multivariate regression analysis was performed to determine the independent association, and higher age at initial chemotherapy and TBI significantly decreased the odds of agenesis. In the regression analysis for microdontia, the older age at initial chemotherapy significantly decreased the odds of microdontia. However, the univariate analysis of root malformation demonstrated no significant predictors.

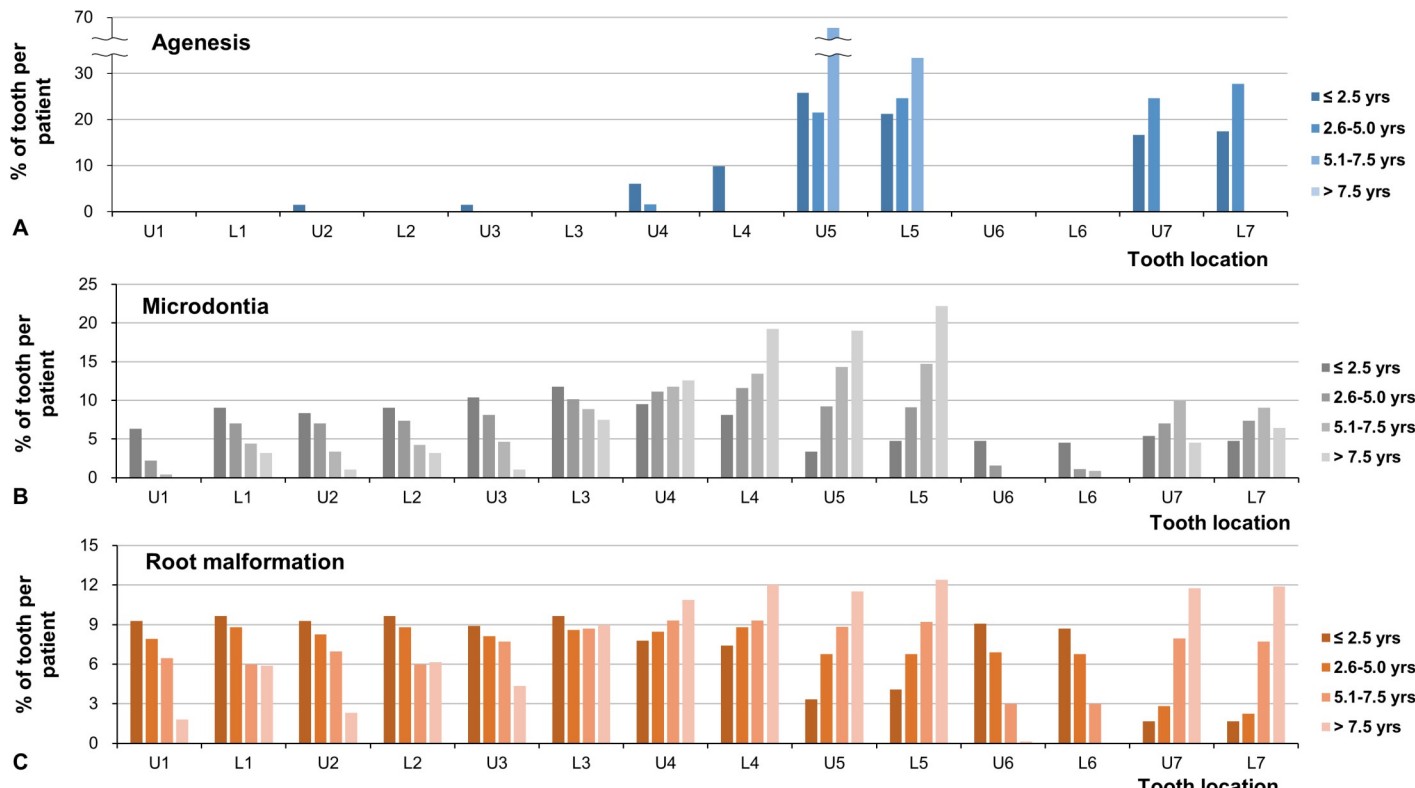

**Fig 4. Distribution of dental anomalies according to the tooth location.** (A) The location of the affected tooth is influenced by the age at chemotherapy depending on the developmental stage of each tooth. The most common tooth exhibiting agenesis is the maxillary second premolar (U5; 25.7%), followed by mandibular second premolar (L5; 22.3%). The incisors (U1-2, L1-2), canines (U3, L3), and first molars (U6, L6) are minimally affected by agenesis and microdontia. Tooth agenesis was present in the 5.1–7.5 age groups, while microdontia and root malformation is found in all age groups. (B) In the younger age groups, microdontia occurred mainly in the anterior teeth (U1-3, L1-3), whereas the premolar area (U4-5, L4-5) and second molar (U7, L7) are mainly affected in the older age groups. (C) Root malformation mainly affected the anterior teeth (U1-3, L1-3) and first molars (U6, L6) in the younger groups, whereas the premolars (U4-5, L4-5) and second molars (U7, L7) are mainly affected in the older groups.

**Table 4. Incidence of dental anomalies according to TBI.**

| Mean number (%) of tooth per patient | ≤ 2.5 | | | | 2.6–5.0 | | | | 5.1–7.5 | | | | > 7.5 | | | |
|---|---|---|---|---|---|---|---|---|---|---|---|---|---|---|---|---|
| | TBI | | non-TBI | P-value | TBI | | non-TBI | P-value | TBI | | non-TBI | P-value | TBI | | non-TBI | P-value |
| Agenesis | 2 | (8.7) | 6 (21.6) | 0.0180* | 1 | (4.3) | 2 (7.9) | NS | 0 | (0.2) | 0 (0.9) | NS | 0 | (0.0) | 0 (0.0) | - |
| Microdontia | 19 | (76.3) | 16 (74.3) | NS | 15 | (58.0) | 18 (69.9) | 0.0458* | 13 | (47.7) | 12 (42.7) | NS | 8 | (29.0) | 7 (24.4) | NS |
| Root malformation | 23 | (89.9) | 20 (91.7) | NS | 23 | (86.1) | 23 (88.0) | NS | 20 | (73.2) | 22 (78.5) | NS | 15 | (54.6) | 16 (57.1) | NS |

Student's *t*-test. NS indicates no statistical significance between the groups.

*P < .05

**P < .01

***P < .001.

These results indicated that earlier age at initial chemotherapy was a risk factors for agenesis and microdontia.

## Discussion

The survival rates of childhood cancer patients have improved greatly and the late side effects of pediatric cancer therapy are becoming increasingly important [1, 2]. Disturbances in dental development have been reported after undergoing hematopoietic SCT combined with chemotherapy and/or radiotherapy [3–6]. Previous reports have mainly focused on the long-term effects of chemotherapy and treatment modalities such as specific chemotherapeutic agents [12, 18, 22] and TBI. Most studies had limitations in that the sample size was too small and long-term dental side effects were not available. To overcome these limitations in the conventional method of sample collection, the concept of big data utilizing a CDW was integrated to optimize the large sample selection and data analysis over a longer time span. Our study examined large cohorts from the CDW of Catholic Medical Center information system (CMC nU)

**Table 5. Univariate and multivariate regression analysis on risk factors for agenesis and microdontia.**

| Types of dental anomaly | Variables | Univariate | | | Multivariate | | |
|---|---|---|---|---|---|---|---|
| | | Odds ratio | 95% confidence interval | P-value | Odds ratio | 95% confidence interval | P-value |
| Agenesis | Age at initial chemotherapy+ | 0.4270 | 0.316–0.577 | 0 < .001*** | 0.4324 | 0.3195–0.5851 | 0.0000*** |
| | TBI | 0.2830 | 0.1180–0.6750 | 0.0040* | 0.2712 | 0.0900–0.8169 | 0.0204* |
| | No. of Chemotherapy agents+ | 0.8450 | 0.6040–1.1800 | NS | | | |
| Microdontia | Age at initial chemotherapy+ | 0.6610 | 0.4680–0.9340 | 0.0190* | 0.6608 | 0.4677–0.9337 | 0.0188* |
| | TBI | 0.4880 | 0.1120–2.1170 | NS | | | |
| | No. of Chemotherapy agents+ | 0.7310 | 0.4230–1.2640 | NS | | | |

NS indicates no statistical significance between the groups.

* P < .05.

** P < .01.

*** P < .001.

+continuous variable.

which has more than five general hospitals and a strong SCT-specialized team. In addition, clinical information regarding the medical and dental aspects of patients undergoing SCT is relatively rare. Therefore, this study was only possible where medical-dental interdisciplinary treatment was active, indicating that many long-term survivors visited dental hospitals after undergoing SCT and had panoramic radiographs taken for dental follow-up.

As the age at initial chemotherapy increased, the prevalence of agenesis decreased, and the number of teeth affected by agenesis per patient also decreased (Table 2 and Fig 3A and 3B). Patients who received initial chemotherapy over the age of 5.0–7.5 years developed an average of 0.168 congenitally missing teeth per patient, whereas patients who received initial chemotherapy at the age of 2.5–5.0 years had a mean number of 1.68 congenitally missing teeth per patient. In addition, the risk of agenesis declined considerably and became negligible when chemotherapy was administered over the age of 5 years. The oldest patient with tooth agenesis was 6.9 years old in this study sample and had three congenitally missing second premolars. No patient had tooth agenesis in the oldest group (> 7.5 years). However, even if chemotherapy was administered relatively late (> 7.5 years old), microdontia (7.3 per patient) and root malformation (15.7 per patients) were unavoidable. The age interval for this grouping was adopted from the Atlas of Human Tooth Development and Eruption [23] with some modifications.

The developmental stage of each tooth is different at each time point [23] as each tooth forms from the crown to the root. Based on the result of this study, the following hypothesis can be formulated (Table 2 and Figs 3 and 4). If the tooth of the patient was at the crown development stage at the time of initial chemotherapy, it was congenitally absent or microdontic. Likewise, if the crown of the tooth has already formed and the root is developing, then the affected tooth will develop a root malformation. This phenomenon may occur according to the location of each tooth depending on the developmental stage. Incisors (U1-2, L1-2), canines (U3, L3), and first molars (U6, L6) developed at a relatively early age after birth, whereas premolars (U4-5, L4-5) and second molars (U7, L7) developed later. Therefore, if chemotherapy is administered relatively early, anterior teeth and the first molars will be affected. Subsequently the affected area will gradually move to the premolars and second molar area region as the age at initial chemotherapy increases. In other words, only the teeth that are developing at the time of chemotherapy may be vulnerable to developmental complications. Although assessments of third molars and deciduous teeth were not included in this study, deciduous teeth and third molars were also affected by agenesis, microdontia, and root malformation in the same way if chemotherapy was administered at younger and older ages, respectively. These results indicated that the location of the affected tooth was influenced by the age at initial chemotherapy, depending on the developmental stage of each tooth. In addition, the high prevalence and large extent of dental developmental complications may translate clinically to early dental examination and timely treatment being advised for long-term survivors who underwent SCT in their early childhood.

The results shown in Table 4 were rather unexpected as the incidence of agenesis was higher in the non-TBI group than that in the TBI group, with a statistical significant difference in the youngest age group (P < 0.05). TBI has adverse effects on dental development [4, 7, 8, 11]. In contrast, Nishimura et al. [18] reported that the incidence of developmental anomalies did not show significant differences between the TBI and non-TBI groups. As the non-TBI group in this study had a higher incidence, especially of agenesis, patients with or without TBI were further categorized by diagnosis, presence of agenesis, and mean age at initial chemotherapy. The timing of initial chemotherapy in the non-TBI group with agenesis was significantly earlier than that in the TBI and agenesis-free groups. This may partly explain why the incidence of agenesis was higher in the non-TBI group. Therefore, TBI may not be interpreted as a risk factor in this study sample.

Concerning the clinical risk factors influencing dental developmental disturbances, Kang et al. [6] reported that cancer diagnosis at ≤ 3 years of age, a history of hematopoietic SCT, the

use of more than four classes of chemotherapeutic agents, and the use of heavy metal agents were significantly associated with the development of dental anomalies. Our study demonstrated that age at initial chemotherapy is significantly associated with occurrence of agenesis and microdontia (Table 5). Other potential risk factors include the intensity of conditioning (myeloablative vs non-myeloablative chemotherapy), dosage of TBI, and characteristics of the chemotherapeutic agent. Due to the complexity of several risk factors, further overall assessment with larger sample size is necessary to determine the underlying factors.

According to a recently published study [24], the prevalence of tooth agenesis was reported to be 6.5% in the Korean general population, except for third molars. However, in patients who underwent SCT, prevalence of agenesis was found in 22.2% of patients in this study. In particular, when patients were classified by the age at initial chemotherapy, the prevalence of agenesis was 73.1, 30.8, 7.9, and 0% according to the age group, respectively. Considering the high prevalence of agenesis in the under-five-year-old group, where agenesis mainly occurs, the occurrence of agenesis independent of SCT is limited, but may act as a confounder. Therefore, further studies with large patients compared to a control group should be needed to minimize bias regarding complexity of several risk factors.

Early detection and timely treatment of dental abnormalities are essential to minimize the oral complications of SCT and improve dental health [9, 25]. As the location and extent of dental developmental anomalies in pediatric patients undergoing SCT can be generally predicted from the results of this study, more accurate explanation of the possible dental sequelae can be provided to patients before undergoing SCT. In addition, the high prevalence and incidence of dental developmental complication may translate clinically; thus meticulous clinical and radiographic dental surveillance is necessary. A multidisciplinary approach involving oncologists, orthodontists, prosthodontists, and other related professionals is required to manage the oral condition of children before, during, and after SCT.

## Conclusions

1. The prevalence and extent of dental anomalies increased when the age at initial chemotherapy was lower.

2. Tooth agenesis after undergoing SCT was most prevalent in the second premolars and second molars. None of the patients had tooth agenesis when the initial age at chemotherapy was over 7.5 years.

3. Microdontia and root malformation occurred mainly in the anterior teeth when the age at initial chemotherapy was low. In contrast, they occurred mainly in the premolars and second molars in the older age group.

4. TBI did not significantly increase the prevalence and extent of dental complications.

5. Based on the age of their initial chemotherapy, the location and extent of dental developmental anomalies in pediatric patients undergoing SCT may be predicted.

These results suggest that careful dental follow-up and timely treatment are recommended for long-term survivors who undergo SCT in their early childhood.

## Acknowledgments

We like to acknowledge Mrs. YJ LEE in organizing the data in such a detailed way and Professor TH Cho for consulting for the data interpretation.

## Author Contributions

**Conceptualization:** Hee Jin Lim, Yoonji Kim.

**Data curation:** Jaehyun Kim, Hee Jin Lim, Ja Hyeong Ku.

**Formal analysis:** Jaehyun Kim, Hee Jin Lim, Ja Hyeong Ku, Yoonji Kim.

**Funding acquisition:** Yoonji Kim.

**Investigation:** Jaehyun Kim, Hee Jin Lim, Ja Hyeong Ku, Yoonji Kim.

**Methodology:** Jaehyun Kim, Hee Jin Lim, Yoonji Kim.

**Supervision:** Yoonji Kim.

**Writing – original draft:** Jaehyun Kim, Hee Jin Lim, Yoonji Kim.

**Writing – review & editing:** Yoon-Ah Kook, Nack-Gyun Chung, Yoonji Kim.

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
