## [Decision Letter · Decision Letter 0]

4 Oct 2022

PONE-D-22-07144Dental developmental complications in pediatric hematopoietic stem cell transplantation patients: A study using CMC clinical data warehousePLOS ONE

Dear Dr. Kim,

Thank you for submitting your manuscript to PLOS ONE. After careful consideration, we feel that it has merit but does not fully meet PLOS ONE’s publication criteria as it currently stands. Therefore, we invite you to submit a revised version of the manuscript that addresses the points raised during the review process.

We look forward to receiving your revised manuscript.

Kind regards,

Nazmul Haque

Academic Editor

PLOS ONE

Journal Requirements:

“This study was supported by Seoul St. Mary’s Hospital, The Catholic University of Korea, Seoul, Korea (ZC20RISI0730 / https://www.cmcseoul.or.kr/en.common.main.main.sp) and National Research Found government (MSIT) (No. 2021R1F1A1049557 / https://www.nrf.re.kr/eng/main/). YK received all award above. All of funders had no role in study design, data collection and analysis, decision to publish, or preparation of the manuscript.”

Additional Editor Comments (if provided):

This manuscript has been written on an interesting topic. However, there are several major issues needed to be resolved before further consideration of this manuscript.

1. All the methods are not properly described in the 'Materials and Methods' section. Methods are written in the result section as well.

2. Discussion is very poorly written. Revise the discussion section to make it more communicative.

3. This manuscript requires major English revision to improve its readability.

Reviewers' comments:

Reviewer's Responses to Questions

**Comments to the Author**

1. Is the manuscript technically sound, and do the data support the conclusions?

Reviewer #1: Yes

2. Has the statistical analysis been performed appropriately and rigorously? 

Reviewer #1: Yes

3. Have the authors made all data underlying the findings in their manuscript fully available?

Reviewer #1: Yes

4. Is the manuscript presented in an intelligible fashion and written in standard English?

Reviewer #1: Yes

5. Review Comments to the Author

Reviewer #1: It´d be greatly to have a control group for further comparative analysis. There is one important question regarding missing congenital teeth. How these teeth were identified amongst the agenesia caused by the SCT?

6. PLOS authors have the option to publish the peer review history of their article (what does this mean?). If published, this will include your full peer review and any attached files.

Reviewer #1: **Yes: **Alexandre Frascino

---

## [Author Response · Author response to Decision Letter 0]

1 Nov 2022

PONE-D-22-07144

Dental developmental complications in pediatric hematopoietic stem cell transplantation patients: A study using CMC clinical data warehouse

Additional requirement 

➜The manuscript had been modified to meet the style requirements of PLOS ONE.

2. Please also include the statement “There was no additional external funding received for this study.” in your updated Funding Statement. Please include your amended Funding Statement within your cover letter. 

➜The Financial Disclosure Statement was modified as guideline by PLOS ONE, and this amended funding statement was included within the revised cover letter.

3. 1) Concerning the Data Availability.

➜The dataset related to this study was uploaded through Figshare, and DOIs were provided in the revised cover letter. 

2) Concerning the minimal data set;

➜This study was conducted in patients after pediatric hematopoietic stem cell transplantation (SCT), we have uploaded all the dataset of 153 eligible subjects instead of the minimal data set for reproducing the results. This large number of study subjects with clinical information of both medical and dental parts of pediatric SCT patients is very rare, and hard-to-find data.

4. We note that you have included the phrase “data not shown” in your manuscript. ➜We have changed the manuscript, and supporting information was provided further in the uploaded dataset.

 Additional Editor Comments (if provided):

1. All the methods are not properly described in the 'Materials and Methods' section. Methods are written in the result section as well. ➜We made changes in the ‘Materials and Methods’ section. Methods in the other section were removed and all the methods are described in proper order. 

2. Discussion is very poorly written. Revise the discussion section to make it more communicative. ➜The manuscript was revised for making it more communicative with the help of professional English editing company. 

3. This manuscript requires major English revision to improve its readability. ➜English revision and proofreading were done by professional editing company (Editage®, https://www.editage.co.kr/). We hope this will improve its readability. 

Reviewers' comments:

Reviewer's Responses to Questions

1. Is the manuscript technically sound, and do the data support the conclusions? Reviewer #1: Yes 

2. Has the statistical analysis been performed appropriately and rigorously? Reviewer #1: Yes

3. Have the authors made all data underlying the findings in their manuscript fully available? Reviewer #1: Yes

4. Is the manuscript presented in an intelligible fashion and written in standard English?

Reviewer #1: Yes

5. Review Comments to the Author

Reviewer #1: It´d be greatly to have a control group for further comparative analysis. There is one important question regarding missing congenital teeth. How these teeth were identified amongst the agenesia caused by the SCT? ➜ Thank you for valuable comments. 

1) It would be hard to identify congenital missing teeth caused or influenced by SCT. However, the prevalence of missing congenital teeth is significantly lower in general population without SCT. According to a recently published study by our group (Ku et al. Common dental anomalies in Korean orthodontic patients: An update. Korean Journal of Orthodontics 2022;52:324-333), the prevalence of tooth agenesis was found to be 6.5% in Korean orthodontic population, not in Korean general population. Majority of patients in orthodontic population are referred from general dentists due to some dental developmental anomalies, therefore, the prevalence of missing congenital teeth in orthodontic population is found to be higher that of general population, which is reported around 5%. (Lee et al. A study on the prevalence of dental anomalies in Korean dental-patients. Korean J Orthod 2011;41:346-53). In this study, prevalence of missing congenital teeth in patients who underwent SCT was found as 22.2%. In particular, when patients were classified by the age at initial chemotherapy, the prevalence of agenesis in the youngest age group (younger than 5 years) was 73.1%. This high percentage rarely occurs in healthy individuals. 

2) In addition to missing teeth, other developmental dental anomalies such as microdontia and root malformation almost always coincides in SCT patients. These unique phenomena are hard to find in healthy individuals with missing teeth. 

3) Therefore, considering the significantly high prevalence of missing teeth and simultaneous dental anomalies the impact on the occurrence of missing teeth unrelated to SCT in this paper would be limited. Further studies with pediatric SCT patients and healthy control group would be ideal to elucidate further.

---

## [Decision Letter · Decision Letter 1]

12 Dec 2022

Dental developmental complications in pediatric hematopoietic stem cell transplantation patients: A study using CMC clinical data warehouse

PONE-D-22-07144R1

Dear Dr. Kim,

We’re pleased to inform you that your manuscript has been judged scientifically suitable for publication and will be formally accepted for publication once it meets all outstanding technical requirements.

Kind regards,

Nazmul Haque

Academic Editor

PLOS ONE

Additional Editor Comments (optional):

Reviewers' comments:

Reviewer's Responses to Questions

**Comments to the Author**

1. If the authors have adequately addressed your comments raised in a previous round of review and you feel that this manuscript is now acceptable for publication, you may indicate that here to bypass the “Comments to the Author” section, enter your conflict of interest statement in the “Confidential to Editor” section, and submit your "Accept" recommendation.

Reviewer #1: All comments have been addressed

2. Is the manuscript technically sound, and do the data support the conclusions?

Reviewer #1: Yes

3. Has the statistical analysis been performed appropriately and rigorously? 

Reviewer #1: Yes

4. Have the authors made all data underlying the findings in their manuscript fully available?

Reviewer #1: Yes

5. Is the manuscript presented in an intelligible fashion and written in standard English?

Reviewer #1: Yes

6. Review Comments to the Author

Reviewer #1: The comments and suggestion met criteria and are eligible for publication.

All the alterations were addressed properly.

7. PLOS authors have the option to publish the peer review history of their article (what does this mean?). If published, this will include your full peer review and any attached files.

Reviewer #1: **Yes: **Alexandre Frascino

---

## [Editor Report · Acceptance letter]

14 Dec 2022

PONE-D-22-07144R1 

Dental developmental complications in pediatric hematopoietic stem cell transplantation patients: A study using CMC clinical data warehouse 

Dear Dr. Kim:

I'm pleased to inform you that your manuscript has been deemed suitable for publication in PLOS ONE. Congratulations! Your manuscript is now with our production department. 

Kind regards, 

on behalf of

Dr. Nazmul Haque 

Academic Editor

PLOS ONE